# Selection of Endophytic *Beauveria bassiana* as a Dual Biocontrol Agent of Tomato Pathogens and Pests

**DOI:** 10.3390/pathogens10101242

**Published:** 2021-09-26

**Authors:** Martina Sinno, Marta Ranesi, Ilaria Di Lelio, Giuseppina Iacomino, Andrea Becchimanzi, Eleonora Barra, Donata Molisso, Francesco Pennacchio, Maria Cristina Digilio, Stefania Vitale, David Turrà, Vili Harizanova, Matteo Lorito, Sheridan Lois Woo

**Affiliations:** 1Department of Agricultural Sciences, University of Naples Federico II, 80055 Portici, Italy; marta.ranesi@unina.it (M.R.); ilaria.dilelio@unina.it (I.D.L.); Giusi.iacomino1992@gmail.com (G.I.); andrea.becchimanzi@unina.it (A.B.); eleonora.barra@unina.it (E.B.); donata.molisso@unina.it (D.M.); f.pennacchio@unina.it (F.P.); digilio@unina.it (M.C.D.); davturra@unina.it (D.T.); lorito@unina.it (M.L.); 2BAT Center-Interuniversity Center for Studies on Bioinspired Agro-Environmental Technology, University of Naples Federico II, 80055 Naples, Italy; woo@unina.it; 3National Research Council, Institute for Sustainable Plant Protection, 80055 Portici, Italy; stefania.vitale@ipsp.cnr.it; 4Department of Entomology, Agricultural University-Plovdiv, 12, 4000 Plovdiv, Bulgaria; vili@au-plovdiv.bg; 5Department of Pharmacy, University of Naples Federico II, 80131 Napoli, Italy; 6Task Force on Microbiome Studies, University of Naples Federico II, 80131 Naples, Italy

**Keywords:** beneficial microbes, *Botrytis cinerea*, *Alternaria alternata*, *Macrosiphum euphorbiae*, *Solanum lycopersicum*, entomopathogenic fungi

## Abstract

Endophytic fungi (EF) can enhance both plant growth and defense barriers against pests and pathogens, contributing to the reduction of chemical pesticides and fertilizers use in agriculture. *Beauveria bassiana* is an entomopathogenic fungus showing endophytism in several crops, often associated with a good capacity to limit the development of pests and disease agents. However, the diversity of the protective efficacy and plant response to different strains can be remarkable and needs to be carefully assessed for the successful and predictable use of these beneficial microorganisms. This study aims to select *B. bassiana* strains able to colonize tomato plants as endophytes as well as to control two important disease agents, *Botrytis cinerea* and *Alternaria alternata,* and the pest aphid, *Macrosiphum euphorbiae*. Nine wild-type isolates and one commercial strain were screened for endophytism, then further characterized for plant-growth promotion plus inhibition of disease development and pest infestation. Four isolates proved to have a good control activity against the biotic stressors tested, but only Bb716 was also able to promote plant growth. This work provides a simple workflow for the selection of beneficial EF, paving the way towards more effective use of *B. bassiana* in Integrate Pest Management (IPM) of tomato.

## 1. Introduction

Feeding the growing world population while decreasing the environmental impact of agriculture is one of the biggest challenges of our time that must be confronted in the present and future [1,2,3]. A major concern for the scientific community is to provide new strategies to guarantee food security plus the safety of agricultural products as well as applications to reduce chemical pesticides and fertilizers [4].

Among the explored alternatives, the use of beneficial microorganisms (BMs) is one of the main pillars to provide a green turn in agriculture farming systems worldwide due to their noteworthy potential to increase crop health and fitness plus limiting negative impacts on the environment [5,6]. The most relevant plant-benefit effects demonstrated by BMs are the control and induced resistance to pest and pathogen attack, increased tolerance to abiotic stress, improved plant nutrition, plus growth promotion [7,8,9,10,11].

In this context, increased attention has been given to the role of fungal endophytes for crop protection, to control the causal agents of disease and infestation, as well as enhance plant fitness [12,13,14,15], and increase above- and/or below-ground biomass associated to improved productivity and yield [12,16,17]. The main advantage of endophytic colonization is attributable to the in-depth association of the host plant and its microbial partner that permits physical-biochemical contact, with the prompt availability of the bioactive molecules produced by the fungus, released within the plant tissues [12,13,14,15,16,17,18]. Many fungal endophytes are known to secrete a plethora of bioactive compounds that largely underlie the positive effects to the plant, which can have a direct antimicrobial or insecticidal effect and/or act indirectly to stimulate the plant defense response or growth [12,19,20,21].

Plant protection exerted by beneficial endophytic fungi (EF) may be direct or indirect, whereby direct processes include parasitism, competition (nutrition, space), and inhibition of microorganisms/pests due to the release of substances, such as antibiotics, toxins, and lytic enzymes [22,23]; and indirect mechanisms embrace the induction of plant defense. Induced Systemic Resistance (ISR) and Systemic Acquired Resistance (SAR) are two plant-defense responses that may be activated by microorganisms, possibly triggering a priming effect in the plant that activates a precocious response to biotic and abiotic stress, that differ according to the metabolic pathways activated and regulated in the plant [24,25,26]. SAR may be activated by pathogen/pest attack and/or elicitor molecules, regulated in the plant by the salicylic acid (SA) pathway associated with the signaling of SAR genes, such as those encoding for pathogenesis-related (PR) proteins [27]. On the other hand, ISR may be induced by colonization of non-pathogenic EF or plant-growth promoting rhizobacteria, and it is mediated by jasmonic acid (JA) and ethylene (ET) pathways [24,25,28,29,30]. Nevertheless, it should be noted that SAR and ISR are two distinct defense mechanisms, but they are not necessarily independent and may overlap due to crosstalk between the hormonal pathways [30,31].

The entomopathogenic fungus, *Beauveria bassiana* (Bals.) Vuill. (Ascomycota: Hypocreales), is largely used as an alternative to chemical pesticides for the biocontrol of insect pests and is the active ingredient of several commercial products used worldwide for sustainable pest management [32,33]. In the last decade, the capability of *B. bassiana* to endophytically colonize a wide range of host plants has been proven as well as its capacity to induce plant resistance against insect pests and pathogens [22,34,35,36,37,38,39,40,41].

*B. bassiana* has been reported to naturally grow as an endophyte in the above and below-ground vegetative tissues, usually acquired by the plant from the surrounding environment by horizontal transmission, although vertical transmission via seeds has been verified in *Papaver somniferum* L. [36,38,42]. Furthermore, it has been demonstrated that *B. bassiana* can be artificially introduced as an endophyte in several plant species by using different inoculation methods, such as seed coating, soil watering, root dipping, and foliar spraying [40,43,44,45,46,47]. In this plant-microbe interaction, there is a reciprocal exchange of benefits: the fungus obtains nutrients, and the host plant receives growth stimulation, enhanced resistance to insect pests, and protection against pathogen attack [40,48,49,50].

Tomato, *Solanum lycopersicum* L., is an important horticultural crop worldwide in which there is a growing demand for its agrifood products, thus requiring an ever-increase in yield as well as a wider area under cultivation [51]. One of the major limiting factors affecting food production is the portion of crop yield loss due to deterioration of the products by pest infestation and diseases during harvest, transport, and conservation [51,52]; thus, the implementation of sustainable strategies to control tomato pests and pathogens is highly desirable. In tomato, some bioassay methods have been successfully developed to obtain endophytic colonization with *B. bassiana* to evaluate its potential as a biocontrol agent (BCA). In a recent review on the EF of tomato, Sinno et al. summarized the methods and the results obtained so far with artificial inoculation to improve the performance of this horticultural crop [16]. It has been reported that endophytic *B. bassiana* may protect tomato from the attack by pathogens, such as *Rhizoctonia solani* Kühn, *Botrytis cinerea* Pers., and *Fusarium oxysporum* Schltdl, plus pests, such as *Aphis gossypii* Glover, *Bemisia tabaci* Gennadius, *Empoasca vitis* Goethe, *Helicoverpa zea* Boddie, *H. armigera* Hübner, *Otiorhynchus sulcatus* Fabricius, *Planococcus ficus* Signoret, *Spodoptera littoralis* Boisduval, *S. exigua* Hübner, and *Tuta absoluta* Meyrick ([16] and references within). Pra-bhukarthikeyan and colleagues indicated an increase of defense-related enzymes and poly-phenols in tomato foliar tissues caused by soil inoculation with *B. bassiana*, which are probably responsible for the induction of plant resistance to harmful biotic agents [53].

The objectives of the present study were to select prospective *B. bassiana* (Bb) strains able to endophytically colonize tomato, then subsequently evaluate their capacity to provide crop protection from attack by two foliar pathogens, *B. cinerea* and *Alternaria alternata* (Fr.) Keissl., and the aphid pest *Macrosiphum euphorbiae* Thomas. Furthermore, other than the biocontrol characteristics, isolates were screened to determine the effects of fungal endophytic colonization on plant biostimulation. Although a growing number of studies have investigated the biocontrol potential of *Beauveria* species, to our knowledge, this is the first study reporting a comprehensive screening that evaluates both dual biocontrol and plant-growth promoting capacities of selected isolates, further expanding the list of target pests and pathogens that can be concurrently controlled by this endophytic fungus to *A. alternata* and *M. euphorbiae*.

## 2. Results

### 2.1. Induction and Assessment of Endophytic Colonization

Overall, the 10 *B. bassiana* isolates used in this study were able to endophytically colonize tomato cv. Dwarf San Marzano two weeks post-inoculum. The colonization rate, indicating the percentage of colonized tomato plants, was highly variable depending upon the plant tissue examined (Figure 1A). Seven out of 10 strains were re-isolated in the roots of 100% of the inoculated plants, whereas in the leaves, the overall rate of colonization ranged from 10 to 50%, and in the stems, this varied from 50 to 100% of the treated plants.

The colonization frequency of *Beauveria* in the different tissues collected from the same plant differed significantly among the tested strains and resulted higher in the roots than in the stems and greater in the stems compared to the leaves (Figure 1B). The colonization frequency observed in the roots and in the stems was significantly higher in tomato plants treated with Bb716, Bb74040, and Bb688 strains (one-way ANOVA: roots, F _(9, 50)_ = 2.249, *p* = 0.033; stems, F _(9, 50)_ = 2.658, *p* = 0.013). In the leaves, it resulted significantly higher in Bb716-treated plants (one-way ANOVA: F _(9, 50)_ = 3.598; *p* = 0.001). Taking into account both the colonization rate and frequency, the isolates Bb74040, Bb688, and Bb716 showed the best ability to endophytically colonize the tomato plants.

### 2.2. Biocontrol of Tomato Foliar Pathogens—Test In Vitro

The plate confrontation assays visibly showed that all tested *B. bassiana* strains were able to inhibit the growth of both *A. alternata* and *B. cinerea* in vitro, demonstrating a clearing zone between the pathogen and the BCA colonies in all the experimental plates (Figure 2).

In the case of *B. cinerea*, the mycelial radial growth of the pathogen was significantly lower in the presence of all 10 tested *Beauveria* strains (one-way ANOVA: F _(10, 33)_ = 35.13, *p* < 0.0001) (Figure 2A), with the width of the inhibition zone variable depending upon the strain, ranging from 2.1 mm with Bb632 to 7.5 mm with Bb688 (one-way ANOVA: F _(9, 30)_ = 42.49, *p* < 0.0001) (Figure 2B). The percent of growth inhibition (PGI%) was also highly variable depending on the tested strain; indeed, it ranged from 21% for Bb709, which was the least effective strain, to 65% for Bb688, which showed the highest biocontrol potential against *B. cinerea* along with Bb762 and Bb632, characterized by growth inhibition of 57% and 52%, respectively (one-way ANOVA: F _(9, 30)_ = 16.05, *p* < 0.0001) (Figure 2C).

Additionally, in the case of *A. alternata*, the radial growth of the pathogen mycelium was always significantly lower when placed in dual culture with *B. bassiana* (one-way ANOVA: F _(10, 33)_ = 72,72, *p* < 0.0001) (Figure 2A). The clearing zone of growth inhibition was always evident except for the Bb632 strain, where it was barely noticeable, but highly variable depending on the tested strain, as it ranged from 1.6 mm for Bb709 to 9.3 mm for Bb716 (one-way ANOVA: F _(9, 30)_ = 82.02, *p* < 0.0001) (Figure 2B). The PGI was also strain-dependent, but it was reduced and less variable in comparison to that observed with *B. cinerea*, ranging from 22 to 40% for all the experimental strains (one-way ANOVA: F _(9, 30)_ = 16.05, *p* < 0.0001) (Figure 2C). The most effective strains in limiting the mycelial growth of *A. alternata* were Bb688 and Bb762, both with a PGI of 40%.

### 2.3. Selection of Potential EF and Biocontrol B. bassiana Strains

The selection of *B. bassiana* strains was based firstly on the ability of the tested isolates to endophytically colonize the tomato plant, then secondly on the biocontrol potential assessed in vitro with dual-confrontation tests with *B. cinerea* and *A. alternata*. Strains demonstrating 100% colonization of the plant roots were selected for further evaluation, and three strains with lower root colonization, such as Bb672, Bb682, and Bb758, were no longer considered (Figure 1A).

Secondly, the most effective strains in counteracting pathogen growth in vitro were selected. The bioassay against *A. alternata* did not show noteworthy differences between the isolates; therefore, the biocontrol of *B. cinerea* was used to exclude the strain Bb709 with the lowest efficacy. Thus, the six most promising strains used thereafter were: Bb74040, Bb632, Bb633, Bb688, Bb716, and Bb762.

### 2.4. Plant Growth Promotion Assay

The overall effect of the *B. bassiana* endophytic colonization on the plant biometric parameters was highly strain-dependent and produced opposing effects. Only the Bb632- and Bb716-treated plants were significantly greater than the untreated control, whereas a negative effect was noted for plants colonized by isolate Bb633 (one-way ANOVA: F _(6, 98)_ = 4.227, *p* = 0.0008) (Figure 3A).

The mean leaf area of the third, fourth, and fifth true leaf from each plant showed no significant differences among five measured strains, whereas a decrease was reported in the Bb633-treated plants in comparison to the control (one-way ANOVA: F _(6, 126)_ = 11.11, *p* < 0.0001) (Figure 3B).

The plant dry weight was significantly higher in Bb632- and Bb716-treated plants, as noted for plant height, while no differences were demonstrated for all the other treatments (one-way ANOVA: F _(6, 98)_ = 5.003, *p* = 0.0002) (Figure 3C). Since strain Bb633 negatively affected plant growth, it was excluded from the subsequent biocontrol assays.

### 2.5. Biocontrol of Tomato Foliar Pathogens—Tests in Planta

The efficiency of endophytic *B. bassiana* to counteract the plant pathogen infections in vivo was variable depending upon the endophytic fungal isolate and the specific phytopathogen. Better effects were observed with the endophytic *B. bassiana* treatments on the biocontrol of *B. cinerea* infection in comparison with *A. alternata*. All strains had similar biocontrol activity, significantly reducing *B. cinerea* infection by 35% compared to the control (one-way ANOVA: F _(6, 84)_ = 8.955, *p* < 0.0001) (Figure 4A). The control of *A. alternata* disease symptoms was variable among the strains, whereby four out of five *B. bassiana* treatments (Bb74040, Bb762, Bb688, and Bb716) produced a substantial reduction in symptoms (Figure 3B), and the best reduction in the tomato leaf spot attack by almost 40% was noted with Bb74040 (one-way ANOVA: F _(6, 84)_ = 6.955, *p* < 0.0001) (Figure 4B).

The area of the necrotic lesions caused by the fungal pathogen infection was substantially reduced in the plants colonized with the *B. bassiana* endophyte. *B. cinerea* symptoms were significantly lower than the control in all instances, with strain Bb762 showing the greatest reduction in disease, almost 70%, relative to the untreated plants (one-way ANOVA: F _(6, 76)_ = 5.954, *p* < 0.0001) (Figure 4C). In the case of *A. alternata,* the foliar damage was significantly lower for all the strains except for Bb632 when compared to the control, with strain Bb74040 showing the strongest biocontrol effect (one-way ANOVA: F _(6, 91)_ = 5.851, *p* < 0.0001) (Figure 4D).

### 2.6. Biocontrol of the Aphid M. euphorbiae—Test in Planta

Aphids that were reared on *B. bassiana*-treated plants for their entire lifespan were negatively affected in terms of survival and fertility in comparison to the control cohorts.

Adults of *M. euphorbiae,* feeding on tomato plants endophytically colonized by *B. bassiana,* showed a significantly lower survival compared to the adults fed on control plants regardless of the EF strain used (log-rank test: χ^2^ = 64.27; *p* = 0.0001; dF = 5) (Figure 5A). There was a clear significant reduction in survival when aphids were reared on plants treated with any of the five *Beauveria* strains in comparison to the control, (log-rank test: Bb74040 − χ^2^ = 42.47, *p* < 0.0001, dF = 1; Bb632 − χ^2^ = 33.38, *p* < 0.0001, dF = 1; Bb762 − χ^2^ = 27.21, *p* < 0.0001, dF = 1; Bb688 − χ^2^ = 32.55, *p* < 0.0001, dF = 1; Bb716 − χ^2^ = 29.93, *p* < 0.0001, dF = 1), which were not significantly different among themselves (log-rank test: *p* > 0.05) (Figure 5A).

Similar effects were also noted in aphid fertility, with a significant reduction observed on *B. bassiana-*treated plants compared to control (one-way ANOVA: F_(5, 72)_ = 16.02, *p* < 0.001) (Figure 5B). Among the strains, the lowest number of newly laid nymphs (offspring of the initial 10 females reared on each plant), was detected in plants treated with Bb74040, in which a decrease of over 70% was shown. This treatment resulted as the most effective EF strain in terms of reducing fertility, and it was significantly different from Bb716, Bb762, and Bb688, but similar to strain Bb632. Dead adult specimens were collected throughout the experiment and among the over 200 cadavers none showed the white muscardine mycosis that is typically provoked by *B. bassiana* infection of the fungus-parasitized insects.

## 3. Discussion

In this work, nine wild-type isolates and one commercial strain of *B. bassiana* were tested for their capacity to colonize tomato cv. Dwarf San Marzano, act as biocontrol agents against *Botrytis* and *Alternaria* pathogens plus the *Macrosiphum* pest, as well as perform as a plant-growth promoter. The procedure developed to select promising BM strains of the entomopathogenic fungus *Beauveria bassiana* as endophytic dual biocontrol agents and plant biostimulators is presented in the experimental workflow design (Figure 6). Initially, the capacity of the fungal strains to internally colonize the plant was assessed. Simultaneously, the ability to inhibit pathogens in vitro was tested. The most effective strains were subjected to in vivo bioassays that simulated the real-life plant–soil–microbe interactions in controlled conditions to determine the effect on plant growth, the control of the two foliar fungal pathogens and the aphid pest in vase experiments. In the end, among the 10 *B. bassiana* strains screened, only four isolates proved to have a noteworthy capacity as dual biocontrol agents providing promising protection to tomato from the harmful biotic agents tested.

The selected strains were Bb74040, isolated from a commercial product, plus three wild-type isolates (Bb762, Bb716, and Bb688), were able to negatively affect the survival and fertility of the sucking insect pest *M. euphorbiae* and to significantly decrease the symptoms caused by two foliar fungal pathogens: *A. alternata* and *B. cinerea*. Moreover, Bb716 strain, in addition to biocontrol, also was able to promote plant growth, significantly increasing plant height and dry weight, demonstrating to be an overall beneficial EF in a global sense (Table 1). These results underlined that strain screening is crucial in order to select valuable and efficient BMs since the diversity of the protective efficacy and plant response to different strains can be remarkable and needs to be carefully assessed for successful and predictable use of *B. bassiana* in agriculture.

The results indicated that tomato plant tissues were endophytically colonized by *B. bassiana* following a repeated soil-drench treatment with a conidia suspension. All of the 10 experimental isolates were able to successfully infect and establish themselves in tomato roots, stems, and leaves. Endophytic colonization observed within the plant tissues was much higher in the hypogeal tissues of the roots, where the colonization rate was close to 100% compared to the leaves, where it ranged from 10 to 50%. Recent studies have revealed that the colonization of *B. bassiana* is not uniform, as it remains concentrated in the zones where the inoculum is applied, even if the translocation towards more distal zones may occur but to a lesser extent [36,41]. In fact, the endophyte spread mainly relies on passive transmission, as it migrates within plant tissues due to water translocation [54,55,56,57]. This pattern was confirmed in the present study, where translocation from roots to leaves was observed for all the tested strains 14 days post-inoculum (dpi). This supports the hypothesis of systemic colonization since the inoculum was at the root level, and the tested fungal strains were isolated from stem and leaves, as reported by numerous studies [23,44,58,59,60]. Further studies should be directed towards defining the endophytic colonization window, in order to understand if by using the current protocols of infection, *B. bassiana* is established as an endophyte until the end of the plant life cycle or if this relationship is just transient.

Furthermore, the results presented have shown that several *B. bassiana* strains had a noteworthy capacity for biocontrol against the grey mold *B. cinerea* and the tomato leaf spot agent *A. alternata* both in vitro and in vivo as endophytes. In vitro, using plate dual-culture test, the presence of a clear zone of growth inhibition between the pathogen and the BCA colonies suggests that the type of interaction between the pathogens and *B. bassiana* may be of antibiosis [61,62]. This effect was already reported against *B. cinerea*, *Cladosporium herbarum* (Pers.) Link, *Fusarium* spp. Link, *Gaeumannomyces graminis* var. *tritici* J. Walker, and *R. solani,* but to our knowledge, this is the first report regarding *A. alternata* [39,63,64,65,66,67,68,69]. The remarkable rate of pathogen growth inhibition implemented by some *B. bassiana* strains used in this study could be linked to the ability of these fungi to produce an abundant variety of bioactive metabolites with antimicrobial properties [70]. This was also confirmed in a recent study demonstrating the antifungal properties of the culture filtrate of *B. bassiana* against various phytopathogens, including *Alternaria tenuis* [71]. The most recognized metabolites produced by *B. bassiana* are oosporein, beauvericin, bassianolide, bassianin, beauveriolide, bassiacridin, and cyclosporine, and among them, beauvericin and oosporein evidenced remarkable antibiotic and antifungal properties and are probably responsible for the fungal growth inhibition observed in the bioassay presented here [39,72,73,74,75,76,77,78].

*In vivo*, four strains, Bb74040, Bb762, Bb716, and Bb688, significantly reduced the symptoms caused by *B. cinerea* and *A. alternata*—with a decrease in the infection rate, respectively ranging from 32 to 40% and 21 to 41% with a reduction of the necrotic area ranging from 35% to 73% and 45% to 63%—and, to our knowledge, this is the first evidence of endophytic *B. bassiana* biocontrol against *A. alternata*. The reduction of disease symptoms in *B. bassiana*-treated plants could be explained by direct or indirect mechanisms linked to the presence of the endophyte within the plant tissues. As already mentioned, the plant colonization was highly concentrated at the root level, whereas the pathogens were inoculated on leaves, thus, the direct interaction between the BCA and the pathogens appears to be unlikely. In particular, one of the most effective strains, Bb762, colonized the leaf tissue only in 30% of the inoculated plants with a frequency of about 20%, but the treated plants were remarkably less affected by both pathogens, thus suggesting that the biocontrol mechanism underlying this effect could be indirect. One important indirect mode of action may be due to the above-mentioned secondary metabolites, which may be active in the plant tissues at a distance from the point that they are produced [79]. This action is well-known to occur in the genus *Trichoderma*, another biocontrol fungus, whereby the application of its active secondary metabolites leads to improvement in plant fitness, growth, and protection from harmful biotic agents [31,80,81,82,83,84]. Another important mechanism of action in the biocontrol effect of endophytic *B. bassiana* may be the induction of plant resistance by ISR or SAR or a combination of both. ISR is an essential process known to be triggered by endophytic BM, by which the plant is primed for improved defense against a wide range of pests and pathogens [25]. One proof for ISR is the reduction of disease incidence in plant parts distant from the location of the inoculated beneficial agent, as observed in the bioassay reported in this work [18]. ISR in tomato plants by endophytic *B. bassiana* has been well documented in the literature [18,41,53,66,85]. Thus, it may be hypothesized that one of the diverse indirect mechanisms of action or a synergistic committance responsible for the observed biocontrol. Nonetheless, the biological processes underlying the biocontrol effect here described remains to be further investigated.

The activity of endophytic *B. bassiana* against the aphid *M. euphorbiae* was noted in all the five strains used in the bioassay, which equally acted as valuable biocontrol agents. Aphids reared for their entire lifespan on *B. bassiana*-colonized plants showed a strong decrease in survival and fertility. Since none of the adult aphid cadavers stored for three weeks showed symptoms of mycosis, it was assumed that this fitness reduction is linked to an indirect effect of the fungal endophytic colonization. One of the possible explanations is that of a toxic or antifeeding effect of the treated plants. In fact, the deterrent and toxic effect of colonized plants against insects was reported more than once in literature and could be again associated with the production of fungal secondary metabolites [47,53,56]. This is thought to be one of the modes by which biocontrol is triggered by endophytic entomopathogenic fungi against pests [18]. On the other hand, another mechanism that several authors reported is once again ISR in which the endophyte induces the activation of plant defenses that suppress insects [14,40]. As evidenced by a recent study, the resistance of tomato plants root-colonized by *Trichoderma* against *M. euphorbiae* is related to an increase of plant endogenous defense processes attested by the up-regulation of the transcripts coding for ET, JA, SA, and PR proteins that are likely responsible for a primed state of the plants, as described for ISR and SAR [86]. It can be hypothesized that a similar mechanism underlies the biocontrol effect that was observed in *B. bassiana*-colonized plants, but further investigation need to be undertaken to shed light on this important aspect. Endophytic *B. bassiana* was reported to control various species of aphids on different crops, such as *Myzus persicae* Sulzer infesting chili pepper [87], *A. gossypii* infesting cotton [88], or *Sitobion avenae* Fabricius infesting maize [89], but to our knowledge, this is the first evidence of endophytic *B. bassiana* controlling *M. euphorbiae* infestation.

Bb74040 strain, isolated from a commercial product, has been already introduced as an endophyte in numerous crops including tomato, but to our knowledge, this is the first report in which the tomato endophytic colonization with this strain was achieved with a soil watering of spore suspension [90]. Its application as an endophyte determines multiple beneficial effects on various crops, such as plant-growth promotion in wheat and fava bean [64,91], reduction of disease incidence by *Fusarium* spp. in sweet pepper and wheat [64,92], Zucchini yellow mosaic virus (ZYMV) in squash [93], and *Plasmopara viticola* (Berkeley and Curtis) Berlese and de Toni in grapevine [94]; it also decreased the infestation of insect pests, such as *P. ficus, E. vitis,* and *O. sulcatus* on grapevine [95,96]; *Delia radicum* L. on cabbage [97]; and *M. persicae* on pepper [98,99]. In recent work, Klieber and Reineke pointed out the endophytic capacity of this strain applied to tomato plants by spraying the solution of the fungal-based commercial product directly on the leaf surface and its activity against the tomato leaf-miner *Tuta absoluta* (Lepidoptera: Gelechiidae) [90]. In conformity with this study, the results presented showed that endophytism by strain Bb74040 induced resistance against pests without affecting plant growth, suggesting that products based on *B. bassiana* are highly versatile and well fit in the context of integrated pest/pathogen management.

Endophytic *B. bassiana* was proposed as a dual control agent in a recent review by Jaber and Ownley that underlined the efficiency of this BM to control both pests and pathogens and to induce systemic resistance in numerous crops [18]. This is corroborated by the findings presented here, indicating that two consecutive soil drenches with *B. bassiana* may protect tomato plants from foliar pathogens and aphid pest attacks; thus, the selected isolates represent amenable candidates for a dual biocontrol strategy. Moreover, a soil-drench treatment may have a direct effect on pests and pathogens inhabiting the rhizosphere, providing extra-protection to plants, and be beneficial also from a nutritional perspective, due to the known capability of endophytic entomopathogens, such as *Metharizium*, to provide nitrogen transfer from insects to plants [100], although these beneficial side-effects of the treatment need further investigation for tomato and *Beauveria*. These aspects are particularly relevant in the framework of integrated pest and pathogen management programs. The fungus may colonize plants and provide protection from a wide range of biotic stress; thus, the use of *B. bassiana* could be considered as a preventive control measure, meeting the needs for the implementation of food safety and environmental protection and conservation. Nonetheless, since *B. bassiana* is a known producer of many secondary metabolites that are possible mycotoxins, it is essential to assess the risk that the use of this fungus as an endophyte may represent for the environment and the health of humans and animals.

Based on the four endophytic-biocontrol fungal isolates selected in this study, future investigations could focus on developing bioformulations containing a combination of the best *B. bassiana* performers for biocontrol of pathogens plus the best biocontrol of pests plus the best plant biostimulant. Furthermore, a microbial consortia consisting of different BMs, such as entomopathogens, other biocontrol agents, mycorrhizae fungi, and plant-growth promoting rhizobacteria with various crop protection and production properties, could be considered by mixing compatible microorganisms to produce a single, multi-purpose agricultural product [101].

## 4. Conclusions

This investigation demonstrated that the endophytic and entomopathogenic fungus *B. bassiana* is a valuable, prospective biocontrol agent against both insect pests or fungal pathogens of tomato and a potential plant biostimulant. This paves the way for the development of a crop-improvement strategy with a single BM application that offers multiple beneficial effects to the plant. The screening process that was developed showed that the biocontrol efficiency and plant response to different strains can be remarkably diverse; thus, a solid workflow procedure for strain selection is highly necessary to carefully chose the most promising plant BM. However, it should be mentioned that the biocontrol efficacy of pathogens and/or pests plus the plant benefits observed in this study may be influenced by the controlled conditions imposed in the experimental design. In fact, abiotic factors, such as temperature, have been noted to significantly affect the important favorable characteristics of diverse strains of *Trichoderma* tested as the biological control agents as well as their culturability [86]. The next phase in the present BM selection process should involve testing in the field, in the “real-world” cultivation environment to determine if the selected isolates maintain their capacity as useful agricultural products. Overall, this work suggests that dual biocontrol exerted by endophytic *B. bassiana* represents a promising tool worthy of consideration when defining new IPM strategies. This application may represent a valuable alternative to the use of chemical pesticides in agriculture that aids in maintaining the biological equilibrium of the agroecosystem.

## 5. Materials and Methods

### 5.1. Fungal Isolates: Origins, Identification, and Culture Methods

Ten different strains of *B. bassiana* (Bb) were used. Nine wild isolates (Bb716, Bb633, Bb672, Bb682, Bb688, Bb709, Bb632, Bb758, Bb762) were received from the Agricultural University of Plovdiv (Bulgaria), while one commercial strain, ATCC 74,040 (hereby referred to as Bb74040), was isolated from the commercial product Naturalis^®^ (CBC S.r.l., Grassobbio, Italy). All the wild isolates were identified as *B. bassiana* from morphological characteristics using the taxonomy keys of Humber [102]. Furthermore, molecular characterization was performed using the transcription elongation factor (TEF-1α) DNA fragment gene and Internal Transcriber Spacer region (ITS). The sequences obtained were compared with those already present in the GenBank database by applying the BLAST software on the National Center for Biotechnology Information website (http://www.ncbi.nlm.nih.gov/BLAST/, accessed on 25 July 2021) and gave 99–100% similarity with *B. bassiana* as the first result.

*Botrytis cinerea* and *Alternaria alternata* strains were isolated from tomato-infested leaves and morphologically identified [103]. These fungal diseases agents were selected for the bioassays since they are common to tomato, and in particular, the two strains are highly pathogenic to tomato cv. Dwarf San Marzano, and the protocol for infection is already well consolidated [104,105]. All the isolates were maintained in 90mm Petri dishes containing Potato Dextrose Agar medium (PDA, HiMedia, Mumbai, India) in the dark at 25 °C.

### 5.2. B. bassiana Conidia Production on Rice

*B. bassiana* conidia of each isolate were obtained as follows: 500 g of parboiled rice were placed in autoclavable bags equipped with breathable filter bands for air exchange (SacO2, Flanders, Belgium) and then autoclaved for 20 min at 121 °C. A starter liquid culture was prepared by adding 10 mycelial plugs (6 mm^2^) from a fresh PDA culture to a 250-mL Erlenmeyer Flask containing 100 mL of sterile Potato Dextrose Broth (PDB, HiMedia, Mumbai, India) and maintained for 72 h at 25 °C in orbital agitation at 120 rpm. Successively, the autoclaved rice was inoculated with the starter culture in a laminar flow hood. The sealed bag containing the inoculated rice was incubated at 25 °C with light for 5–7 days, until white conidia covered the rice. Conidia were harvested by washing the rice with 500 mL of sterile water under sterile conditions. Conidia concentration was determined by making a dilution series and counting the number of conidia in a hemocytometer (Neubauer counting chamber) under a microscope.

### 5.3. Induction and Assessment of Endophytic Colonization

Tomato seeds were surface sterilized in 1% NaOCl for 5 min, then rinsed 3 times with sterile distilled water to get rid of any surface fungal contamination. Seeds were germinated on Whatman^®^ sterile filter paper (Sigma-Aldrich, Darmstadt, Germany) soaked with sterile distilled water in the dark in an environmental chamber at 25 °C. Seedlings were individually transplanted to 8-cm diameter pots containing 200 mL of sterile commercial soil (Universal Potting Soil, Floragard, Oldenburg, Germany) and kept in a growth chamber at 25 ± 2 °C, 70 ± 10% RH, and photoperiod of 14:10 h light:dark and arranged in a randomized design with 15 plants divided in 3 replicates for each isolate. After one week, the emerged tomato seedlings were watered with a 20 mL of the *B. bassiana* conidial suspension (~1 × 10^7^ conidia/mL) in order to obtain a final concentration of 1 × 10^6^ conidia/mL of soil volume. Control plants were watered with sterile water. The same treatment was repeated after one week.

To determine the endophytic colonization, 5 tomato plants for each treatment were randomly chosen, uprooted, and dissected 2 weeks after the second watering treatment (one-month-old plants), and the surface-sterilized tissues were plated. The roots were carefully washed under tap water before the sterilization process in order to remove the soil. Three leaves for each plant and the stem were surface-sterilized in 1% NaOCl for 3 min, while the roots were maintained in the sterilizing solution for 5 min; after that, all the plant tissues were rinsed 3 times with sterile distilled water. The success of the disinfection procedure was assessed by plating three replicates of 100 µL each of the residual rinse water on PDA medium plates. Each plant tissue was dried on sterile paper, cut into 5 pieces with a sterile scalpel, and pieces were placed on 90 mm Petri plates containing PDA supplied with 1% lactic acid to avoid bacterial contamination. Plates were incubated at 25 °C in the dark and daily monitored to verify the fungal growth emerging from the cut plant tissues.

The endophytic colonization rate for each strain was calculated as the percentage of colonized plants on the total number of screened plants, and the frequency of colonization, as a measure of colonization intensity, was calculated as the percentage of colonized tissue section per plant on the total number of tissue sections [106]. Colonization rate and frequency were assessed separately for root, leaf, and stem tissues.

The fungal mycelia were isolated from the substrate close to the plant tissue and transferred on new plates containing PDA in order to obtain pure cultures for the morphological identification of the isolated fungus. Endophytic *B. bassiana* colonization was confirmed within 14 days after the tissue plating. The experiment was conducted twice.

### 5.4. Biocontrol of Tomato Foliar Pathogens—Tests In Vitro

Plate confrontation bioassays were performed with all experimental *B. bassiana* isolates to determine their potential as BCA against *B. cinerea* and *A. alternata* and to select the most effective ones. A fungal plug, about 6-mm diameter, was obtained from the periphery of fresh mycelial plate cultures for each plant pathogen and beneficial fungi. One pathogen and one *B. bassiana* isolate plugs were placed at approximately 4 cm of distance, in the middle of 90-mm diameter Petri dishes containing one-fifth strength PDA. The *B. bassiana* plugs were inoculated 2 days before the pathogens since its growth is considerably slower in comparison with one of two pathogens, and the simultaneous inoculum may hide the significance of the inhibition effect. Control consisted of plates inoculated only with one of the pathogens. The radial growth of the pathogen colonies, calculated as the mean distance between the inoculation point (center of the colony) and the edge in the four main directions, was measured at 7 dpi.

The biological activity of the beneficial fungi was assessed by comparing the growth of the pathogens with and without the presence of the *B. bassiana* isolates and reporting the presence of an antagonist/inhibition clearing zone formed between the two confronted fungal colonies. The distance of this zone between the pathogen and the beneficial fungi was measured. Fungal mycelial growth was determined by measuring colony radius and the percentage growth inhibition (PGI) of the pathogens was calculated according to the formula: PGI = (C − T) × 100, where C is the radial growth of the pathogen in control plates (mm), and T is radial growth of the pathogen in presence of the antagonist (mm) [107]. Three plates for each combination were prepared, and the bioassay was replicated twice.

### 5.5. B. bassiana Strain Choice for In Vivo Assay

All the assays described below were carried out on 30-da-old tomato plants endophytically colonized as described above, except for the insect bioassay, which was realized using 15-day-old plants.

The *B. bassiana* strains, selected for their ability to endophytically colonize the tomato plant and to counteract the growth of pathogens in vitro, were used for plants colonization, with water served as negative control. The six strains were: Bb74040, Bb632, Bb633, Bb762, Bb716, and Bb688.

### 5.6. Plant-Growth Promotion Assay

To determine the effect of the endophytic colonization with *B. bassiana* on plant growth, a survey of biometric parameters was carried out on Bb-colonized plants. Five plants per treatment were used to evaluate the effect of the endophytic infection on plant growth. The third, fourth, and fifth true leaf of every plant was gently removed, laid on a white paper sheet beside a measuring stick, and shot. The pictures were analyzed using ImageJ, an open-source image processing software [108], to measure the leaf area for each plant. To record the plant weight, each plant was uprooted, and the roots were gently washed under running water to remove the soil. Successively, the entire plant was placed into a paper bag already containing the removed leaves used to measure the leaf area and the bags were placed into an oven at 70 °C for 72 h. When completely dried, the plant was weighed. The experiment was repeated three times.

### 5.7. Biocontrol of Tomato Foliar Pathogens—Tests in Planta

To determine the effect of Bb endophytic colonization on foliar pathogens infection, biocontrol assays against *A. alternata* and *B. cinerea* were conducted. Ten plants per treatment were placed in a high-humidity chamber (80–90% RH, 20 °C) and inoculated with *B. cinerea*, while an additional subset of plants, again 10 per treatment, were placed in a high-humidity chamber and inoculated with *A. alternata* (80–90% RH, 27 °C). Three true leaves for each plant were inoculated with the pathogen conidial suspension laying on the leaf surface a 10 μL-drop of spore suspension at a concentration of 5 × 10^6^ spores/mL. The inoculation point was marked with a permanent marker. The severity of the disease symptoms and the infection progression were evaluated 7 days dpi. The infection rate was calculated as the number of necrosis observed near the inoculation point over the total number of drops. Furthermore, the necrotic area provoked by the pathogens near the inoculation point was measured as follows: the leaves marked with the permanent marker were removed from the plant, laid on a white paper sheet beside a measuring stick, and shot. The pictures were analyzed using the ImageJ software to measure the mean necrotic area for each plant. The experiment was repeated three times.

### 5.8. Insect Rearing

The aphid *Macrosiphum euphorbiae* was collected from an infested field of tomato crops (Battipaglia, Salerno, Italy) and is permanently reared on *S. lycopersicum* plants (cv. Dwarf San Marzano) placed a growth chamber at 23 ± 2 °C, 70 ± 10% RH, and photoperiod of 14:10 h light:dark. This insect pest was chosen since it has been successfully utilized in previous studies in which it was found to readily feed on the tomato cv. Dwarf San Marzano, and the protocol of infestation and monitoring was already well defined [86,109].

### 5.9. Biocontrol of the Aphid M. euphorbiae—Tests in Planta

To assess the effects of endophytic colonization of tomato with *B. bassiana* on sucking insects infestation, a bioassay on the aphid pest *M. euphorbiae* was carried out. The biocontrol assays against *M. euphorbiae* were conducted in a growth chamber at 25 ± 1 °C, 70 ± 10% RH, and photoperiod of 14:10 h light:dark. For each treatment, four plants at the stage of the second true fully expanded leaf were placed in an entomological cage of 35 × 35 × 60 cm in transparent synthetic canvas (Omnes ed Artes s.a.s., Bergamo, Italy) and infested with *M. euphorbiae* as follows: 15 apterous adults were placed on each plant using a fine-tipped paintbrush and allowed to give birth; after 6 h, the adults were removed, and the first instar nymphs were counted. Nymphs’ development was monitored until reaching the adult stage, and at that moment, only 10 adults were left per plant, removing the exceeding ones.

The survival and the fertility of these adult specimens, reared for their entire lifespan on the experimental plants, were observed for 14 days. Three times a week, the number of alive adults on each plant was recorded; moreover, the laid nymphs were counted and removed. When observed, the adult cadavers were removed, stored in groups of 30 specimens for 30 days in 50-mL falcon tubes with holes for air exchange, and checked weekly to observe putative mycosis occurrence under a microscope. The experiment was replicated three times.

### 5.10. Data Analyses

Data were analyzed by one-way Analysis of Variance (one-way ANOVA) and a-posteriori LSD Fisher test using Minitab18 ^®^ software (Minitab, State College, PA, USA). Data were tested for normality with the Shapiro–Wilk test and homoscedasticity with Levene’s test using PAST 3^®^ software (Softpedia, Carlsbad, CA, USA). Survival curves of *M. euphorbiae* were compared by using Kaplan–Meier and log-rank analysis using GraphPad Prism 5.0 software (GraphPad Software, La Jolla, CA, USA). The graphs were made using GraphPad Prism 5.0 software.

## Figures and Tables

**Figure 1 pathogens-10-01242-f001:**
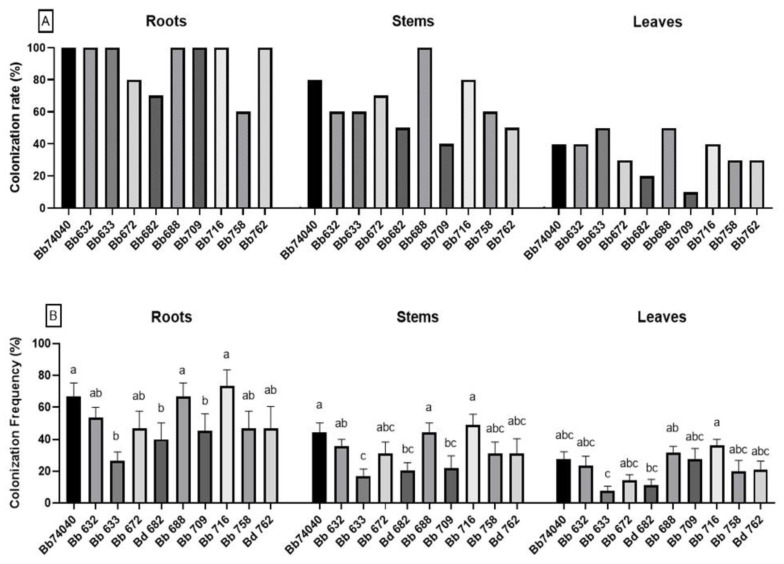
Endophytic colonization of tomato, cv. Dwarf San Marzano, with 10 *B. bassiana* isolates, indicating: (**A**) endophytic colonization rate (%) of all *Beauveria*-treated plants exhibiting fungal growth in the diverse tissues; and (**B**) colonization frequency (%) (means ± SE), which is the percentage of cut tissue sections from the same plant that demonstrate the presence of *Beauveria* mycelium. Bars marked with different letters are significantly different at *p* = 0.05 (Fisher LSD post-hoc test after one-way ANOVA).

**Figure 2 pathogens-10-01242-f002:**
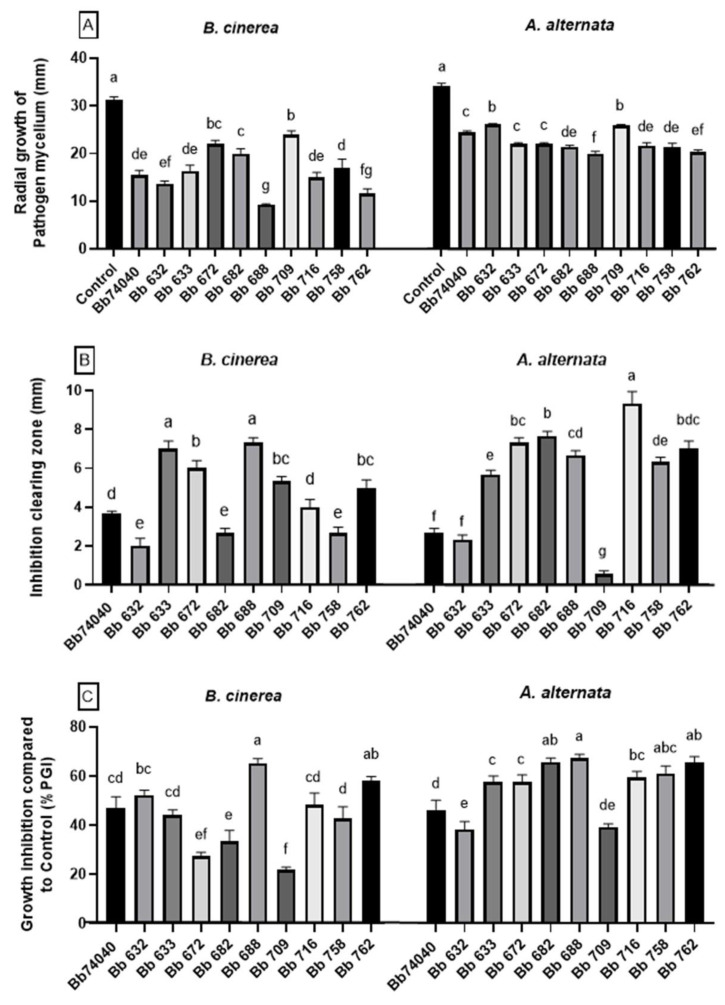
Effect of *B. bassiana* strains on the mycelial growth of fungal pathogens *B. cinerea* and *A. alternata* as noted in in vitro plate confrontation assays (dual-culture tests). (**A**) Radial growth of pathogen mycelium (means ± SE). (**B**) Inhibition clearing zone distance between pathogen and Bb (means ± SE). (**C**) Growth inhibition compared to control (%PGI) (means ± SE). Bars marked with different letters are significantly different at *p* = 0.05 (Fisher LSD post-hoc test after one-way ANOVA).

**Figure 3 pathogens-10-01242-f003:**
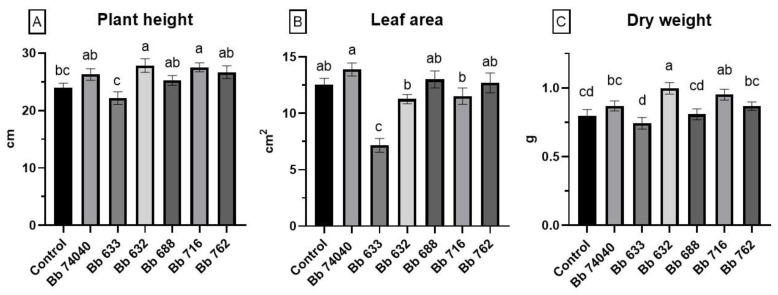
Effect of endophytic colonization with six different *B. bassiana* strains on plant growth of tomato cv. Dwarf San Marzano two weeks post-inoculum. (**A**) Plant height (mean cm ± SE). (**B**) Leaf area of the third, fourth, and fifth true leaf of each plant (mean cm^2^ ± SE). (**C**) Total plant dry weight (mean g ± SE). Bars marked with different letters are significantly different at *p* = 0.05 (Fisher LSD post-hoc test after one-way ANOVA).

**Figure 4 pathogens-10-01242-f004:**
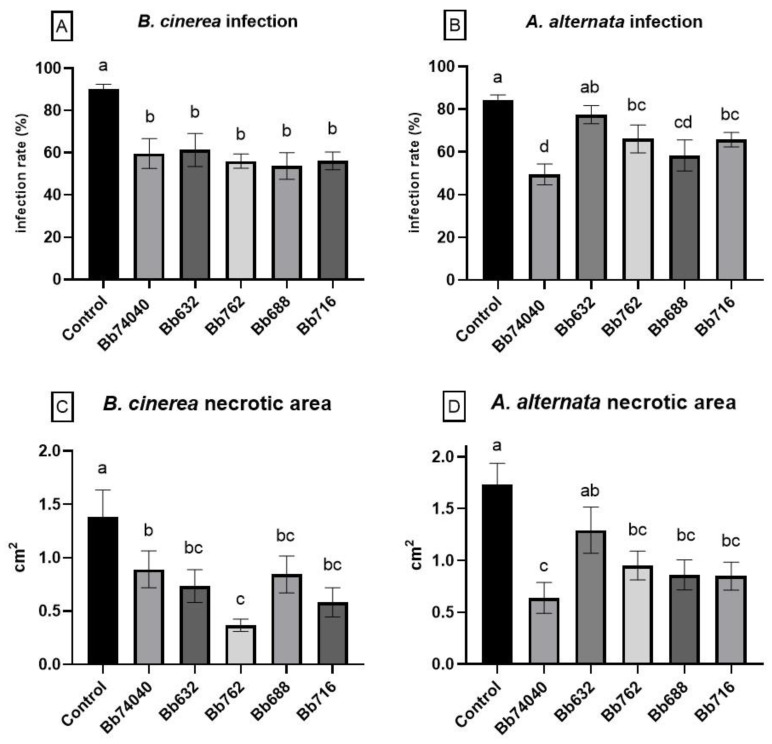
Biocontrol potential of five selected *B. bassiana* isolates against the foliar phytopathogens *B. cinerea* (gray mold) and *A. alternata* (tomato leaf spot) on endophytically colonized tomato plants. Infection rate and percentage of disease symptoms caused by point inoculation with (**A**) *B. cinerea* (mean ± SE) and (**B**) *A. alternata* (mean ± SE); necrotic area, leaf lesions development on leaves inoculated and infected by (**C**) *B. cinerea* (mean ± SE) and (**D**) *A. alternata* (mean ± SE). Bars indicated by different letters are significantly different at *p* = 0.05 (Fisher LSD post-hoc test after one-way ANOVA).

**Figure 5 pathogens-10-01242-f005:**
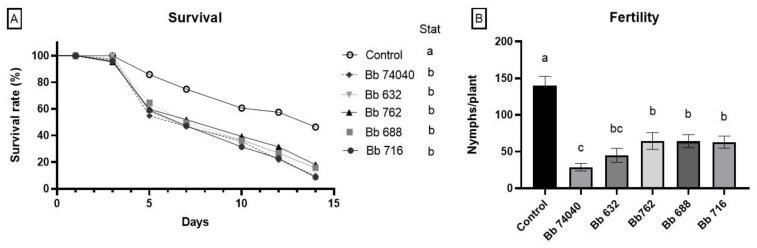
Effects on the aphid *M. euphorbiae* reared for their entire lifespan on plants endophytically colonized by five different isolates of *B. bassiana* or the control plants. (**A**) Survival rate, 14 days after being reared on *B. bassiana*-treated and untreated plants. Different letters in the legend indicate a significant difference (log-rank test, *p* < 0.001). (**B**) Fertility, indicating the number of offspring produced by 10 *M. euphorbiae* adult specimens on each plant during 14 days (means ± SE). Bars indicated by different letters are significantly different at *p* = 0.05 (Fisher LSD post-hoc test after one-way ANOVA).

**Figure 6 pathogens-10-01242-f006:**
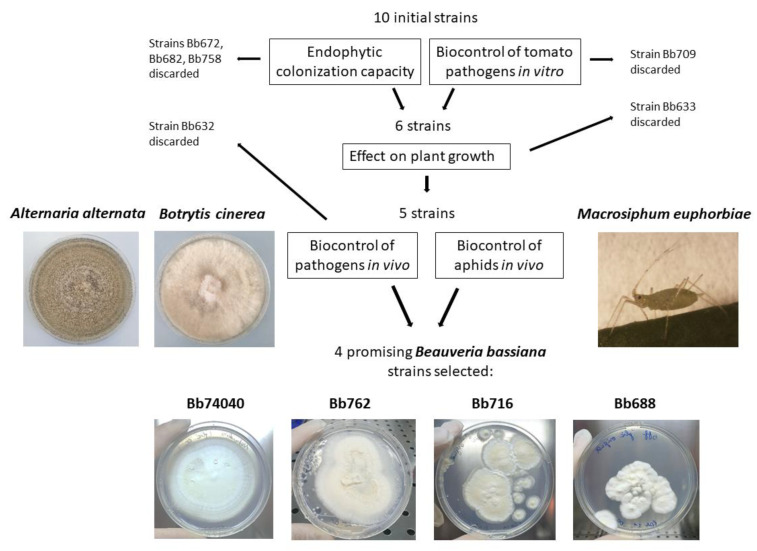
Workflow for the selection of promising BMs belonging to *B. bassiana* selected in terms of capacity for endophytic colonization, plant-growth promotion, and biocontrol of pathogens and pests of tomato.

**Table 1 pathogens-10-01242-t001:** Summary of the overall biological activity of the five *B. bassiana* isolates: biocontrol effect against the pest *M. euphorbiae* and the pathogens *A. alternata* and *B. cinerea* and stimulation effect on plant growth. Biocontrol was evaluated on a scale of four, whereby “0” = no effect, “++” = decrease of the infestation or symptoms of infection, to “+++” = strong decrease of the infestation or symptoms of infection; and the plant-growth effect was evaluated, whereby “0” = no effect and “+” = positive stimulation effect.

*B. bassiana* Isolate	Effect on the Plant Growth	Biocontrol of Tomato Pest	Biocontrol of Tomato Pathogens	
		*Macrosiphum euphorbiae*	*Alternaria alternata*	*Botrytis cinerea*
Bb74040	0	+++	+++	++
Bb632	+	+++	0	++
Bb688	0	+++	++	++
Bb716	+	+++	++	++
Bb762	0	+++	++	+++

## Data Availability

Data is available upon request from the authors; the data that support the findings of this study are available from the corresponding author upon reasonable request.

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
