# Peer review of "Selection of Endophytic Beauveria bassiana as a Dual Biocontrol Agent of Tomato Pathogens and Pests"

_pathogens, 2021, doi:10.3390/pathogens10101242_

Round 1
Reviewer 1 Report
Beauvaria bassiana has been a promising biological control organism that has rarely lived up to expectations in the field. Its elusive success has been frustrating for many. The paper adds to the literature and deserves publication, but I have a few concerns.
1) Efficacy is often related to environmental conditions. The methodology involves nearly ideal conditions for B. bassiana and the pathogens to grow. However, producers and practitioners seldom have such ideal conditions. The pathogens may be more robust than the beneficial organism at lower field temperatures. The methodology should, at the very least, give the reasons for the selection of environmental conditions and acknowledge that results depend on those conditions.
2) Biological control based on a single strain of a single species of beneficial almost never provides results comparable to chemical control, and as a result looks unfavorable. Moreover, a single-strain approach is more likely to select for resistant pathogens. Biological control tends to be more successful when part of an integrated agroecological approach involving multiple tactics. At the very least, the methods section should explain why the ten different strains / isolates were selected, and why a mix of strains were not cultured.
The results tables are sometimes difficult to read.
The English can be improved. The passive voice is used much too often for my taste, but that is a style choice. I won't offer line-by-line editing advice but here are a few examples:
Line 275 "In this work, it was developed a procedure . . ." I suggest "A procedure was developed . . ." The passive voice makes it ambiguous, and an active voice third-person construction would be "The researchers developed . . . "
Line 308 "In this study, B.bassiana was shown to have a noteworthy ability to endophytically colonize the tomato plant tissues . . ." A better passive voice construction would be "Tomato plant tissues were endophytically colonized by B. bassiana in the experiment. I would prefer "The experimental results showed that tomato plant tissues were endophytically colonized by B. bassiana."
Lines 424-427 is a long run-on sentence. Consider making it two or three sentences.
Line 433, "In this investigation, it was demonstrated . . ." Consider "This investigation demonstrated . . ."
Author Response
Response to Reviewer 1 Comments
Beauvaria bassiana has been a promising biological control organism that has rarely lived up to expectations in the field. Its elusive success has been frustrating for many. The paper adds to the literature and deserves publication, but I have a few concerns.
The authors appreciate the Reviewers recognition of the novelty of the research.
Point-by-point revisions, indicating where the changes were made in text:
- Efficacy is often related to environmental conditions. The methodology involves nearly ideal conditions for bassianaand the pathogens to grow. However, producers and practitioners seldom have such ideal conditions. The pathogens may be more robust than the beneficial organism at lower field temperatures. The methodology should, at the very least, give the reasons for the selection of environmental conditions and acknowledge that results depend on those conditions.
- Comments were taken into consideration and this phrase was added to Conclusions – Lines 444-451 “However, it should be mentioned that the biocontrol efficacy of pathogens and/or pests, plus the plant benefits observed in this study may be influenced by the controlled conditions imposed in the experimental design. In fact, abiotic factors such as temperature have been noted to significantly affect the important favorable characteristics of diverse strains of Trichoderma tested as the biological control agents, as well as their culturability [86]. The next phase in the present BM selection process should involve testing in the field, in the “real-world” cultivation environment to determine if the selected isolates maintain their capacity as useful agricultural products.”
- Biological control based on a single strain of a single species of beneficial almost never provides results comparable to chemical control, and as a result looks unfavorable. Moreover, a single-strain approach is more likely to select for resistant pathogens. Biological control tends to be more successful when part of an integrated agroecological approach involving multiple tactics. At the very least, the methods section should explain why the ten different strains / isolates were selected, and why a mix of strains were not cultured.
- Comments were taken into consideration and this phrase was added to Discussion- Lines 427-434
“Based on the five endophytic-biocontrol fungal isolates selected in this study, fu-ture investigations could focus on developing bioformulations containing a combina-tion of the best B. bassiana performers for biocontrol of pathogens plus the best biocon-trol of pests plus the best plant biostimulant. Furthermore, a microbial consortia con-sisting of different BMs, such as entomopathogens, biocontrol agents, mycorrhize fun-gi, plant growth promoting rhizobacteria with various crop protection and production properties could considered, by mixing compatible microorganisms to produce a single multi-purpose agricultural product [106].”
- The results tables are sometimes difficult to read.
- It is not clear what the Reviewers comments are referring to; there is only Table 1 presented.
- The English can be improved. The passive voice is used much too often for my taste, but that is a style choice. I won't offer line-by-line editing advice but here are a few examples: etc.
- Specifically, revisions were made to the examples as indicated by the Reviewer:
- Line 275: “it was developed a procedure” was amended with “A procedure was developed”.
- Line 308: the sentence was revised as asked.
- Lines 424-427: the sentence was split in two.
- Line 433: “In this investigation, it was demonstrated” was amended with “This investigation demonstrated”.
- Additional revisions were made to the English style throughout the text (see Tracking changes).
Reviewer 2 Report
It is a very interesting review where the researchers selected B. bassiana strains able to colonize tomato plants as endophytes, as well as to control some disease and a pest. It is very well organized and explained.
There are only some mistakes:
Keyword: Write the Latin names in italics.
The authority for a Latin binomial name should be provided after each common name the first time it is referred to in the title, abstract, main body, and a figure or table description.
Why have they selected these pathogens and this pest? Explain them in the introduction or add some information about them.
Line 87: When you write some information about tomato, I suggest writing its Latin name.
Line 105: “B. bassiana (Bb)” Why this abbreviation? In the text, it is not written. I think cancel it.
Lines 110-117: They are results or conclusions, not some aims.
Line 125: Write 10 in letter or homogenize all the text either in letter or number format.
Improve the quality of all figures. The letters are pixelated.
Figure 1: Add the different letters and the SE on the bars in the Fig. 1A.
Figure 3: I recommend putting all the figures in a line. And the Fig 3C, correct the unit, it is not “gr”, in the international system is “g”.
Figure 4: I recommend putting all the figures in a line or in two lines.
Line 276, 294, 295, 296: Use short form of Beauveria, Botrytis, Alternaria and Macrosiphum.
Line 332: “Gaeumannomyces graminis var. tritici”, Tritici is in italics.
Line 463: Change grams for the unit “g”.
Lines 557-557: Change HR for RH.
References: Adapt it according to the standards of the journal.
Author Response
Response to Reviewer 2 Comments
It is a very interesting review where the researchers selected B. bassiana strains able to colonize tomato plants as endophytes, as well as to control some disease and a pest. It is very well organized and explained.
Thank you for appreciating our work.
Point-by-point revisions have been made, indicating where the changes were made in text.
- Keywords: Write the Latin names in italics.
- Keywords: the Latin names were written in italics.
- The authority for a Latin binomial name should be provided after each common name the first time it is referred to in the title, abstract, main body, and a figure or table description.
- All of the authorities for the scientific names were added where missing
- Why have they selected these pathogens and this pest? Explain them in the introduction or add some information about them.
- The reason why these pathogens and this pest were selected was added in the materials and methods section.
- Line 87: When you write some information about tomato, I suggest writing its Latin name.
- The Latin name was added.
- Line 105: “ bassiana(Bb)” Why this abbreviation? In the text, it is not written. I think cancel it.
- The abbreviation “Bb” is used more than once in the text so we think it should remain. The abbreviation is written in the Introduction (Objectives) and it is necessary for reference to the isolates in the text and the Figures/Table.
- Lines 110-117: They are results or conclusions, not some aims.
- Agreed – This section removed from the Introduction and integrated into the beginning of the Discussion (lines 291-299)
- Line 125: Write 10 in letter or homogenize all the text either in letter or number format.
- Improve the quality of all figures. The letters are pixelated.
- Modifications to the figures will be determined once the versions from the typographer have been provided.
- Figure 1: Add the different letters and the SE on the bars in the Fig. 1A.
- No statistical analysis was performed for data shown in Fig. 1A, that’s why there are no letters or error bars. It only demonstrates the percentage of plants that were endophytically colonized by bassiana strains.
- Figure 3: I recommend putting all the figures in a line. And the Fig 3C, correct the unit, it is not “gr”, in the international system is “g”.
- Figure 3 is already formatted linearly.
- Figure 4: I recommend putting all the figures in a line or in two lines.
- Not done since the graphs appear too crowded, and too small to read.
- Line 276, 294, 295, 296: Use short form of Beauveria, Botrytis, Alternaria and
- Line 332: “Gaeumannomyces graminis tritici”, Tritici is in italics.
- Done - tritici was corrected and written in italics.
- Line 463: Change grams for the unit “g”.
- Lines 557-557: Change HR for RH.
- References: Adapt it according to the standards of the journal.
- Journal format followed.